# Reinforcement Learning in Many-Agent Settings Under Partial Observability

**Keyang He**[1]        **Prashant Doshi**[1]        **Bikramjit Banerjee**[2]

[1] THINC Lab, Department of Computer Science, University of Georgia, Athens, GA, USA
[2] School of Computing Sciences and Engineering, University of Southern Mississippi, Hattiesburg, MS, USA

## Abstract

Recent renewed interest in multi-agent reinforcement learning (MARL) has generated an impressive array of techniques that leverage deep RL, primarily actor-critic architectures, and can be applied to a limited range of settings in terms of observability and communication. However, a continuing limitation of much of this work is the curse of dimensionality when it comes to representations based on joint actions, which grow exponentially with the number of agents. In this paper, we squarely focus on this challenge of scalability. We apply the key insight of action anonymity to a recently presented actor-critic based MARL algorithm, interactive A2C. We introduce a Dirichlet-multinomial model for maintaining beliefs over the agent population when agents' actions are not perfectly observable. We show that the posterior is a mixture of Dirichlet distributions that we approximate as a single component for tractability. We also show that the prediction accuracy of this method increases with more agents. Finally we show empirically that our method can learn optimal behaviors in two recently introduced pragmatic domains with large agent population, and demonstrates robustness in partially observable environments.

## 1 INTRODUCTION

Continued interest in multi-agent reinforcement learning (MARL) has yielded a variety of algorithms over the years, from Minmax-Q [Littman, 1994] and Nash-Q [Hu and Wellman, 2003] during its initial study to the more recent ones such as MADDPG [Lowe et al., 2017], COMA [Foerster et al., 2018], QMIX [Rashid et al., 2018] and IA2C [He et al., 2021]. While the early methods mostly generalized Q-learning [Watkins, 1992] to multiagent settings, the later

methods utilize the actor-critic schema with centralized or decentralized actor and critic components. The neural network representations of the actor and critic components allow these methods, which by default target settings with perfectly observed states, to expand to partial observability by maintaining a moving window of past observations. While these methods have demonstrated good performance on the standard MARL problem domains, the RL does not practically scale beyond a handful of interacting agents.

Multiagent planning frameworks such as DEC- and I-POMDPs [Bernstein et al., 2002, Gmytrasiewicz and Doshi, 2005] faced a similar hurdle of scaling in a meaningful way to many agents. A key insight – that many domains exhibit the *action anonymity* structure [Jovanovic and Rosenthal, 1988] – helped mitigate this curse of dimensionality (due to many agents) afflicting planning. More specifically, it is the number of agents that perform the various actions – this count vector is called an *action configuration* – which matters to the reasoning rather than the respective identities of the agents performing the actions. In other words, joint action permutations are equivalent. Modeling this invariance enables the planning complexity to drop from being exponential in the number of agents to polynomial thereby facilitating multiagent planning for thousands of agents [Varakantham et al., 2014, Sonu et al., 2017].

Another challenge is to infer actions of other agents in a partially observable setting. Recent methods either assume perfect access to other agents' policies [Yang et al., 2018, Foerster et al., 2018], or infer other agents' policies using their state-action trajectories [Lowe et al., 2017] by maximum likelihood estimation (MLE). While this is appropriate in fully observable environments, in environments where other agents' behaviors can only be partially observed, MLE fails to accurately predict their actions. IA2C [He et al., 2021] manages this uncertainty by maintaining beliefs over candidate models of other agents and updating these beliefs using the noisy observations. However, beliefs must be updated over the models of each other agent, which in general scales exponentially with the number of agents and becomes

*Accepted for the 38th Conference on Uncertainty in Artificial Intelligence* (UAI 2022).

intractable for a large population of agents.

In this paper, we aim to bring the idea of action anonymity to MARL and scale the learning to many-agent settings under partial observability.

1. Our first contribution is to introduce a MARL method for cooperative-competitive settings that integrates action anonymity with the actor-critic learning paradigm. The method tracks the probability distribution over the possible action configurations of other agents.

2. Our second contribution is to replace this distribution over a polynomially-growing space with a Dirichlet-multinomial model over the fixed set of others' actions. We show that the prediction accuracy of this model improves with more agents. However, the noisy observation leads to a posterior that is a mixture of Dirichlets, which we subsequently replace by a single Dirichlet function. This method efficiently updates the Dirichlet using the agent's noisy observations, and achieves an accuracy that is comparable to the traditional belief update procedure. Additionally, it can be implemented by neural network layers and conveniently integrated into most actor-critic based MARL methods to model large agent populations in partially observable environments.

Our experiments demonstrate scalability of the method on two large domains with up to one hundred agents. Moreover, in partially observable environments where states and other agents' actions are not perfectly observed but with noise, this method maintains distributions that accurately predict actions of the agent population, which eventually leads to performance that improves on the previous MARL techniques in both domains.

## 2 BACKGROUND

We briefly describe a recent partially observable cooperative-competitive multi-agent domain in the next subsection, and follow it up with a review of the recently introduced *interactive A2C* (IA2C) method for learning in such settings.

### 2.1 ORGANIZATION DOMAIN

The Organization domain (Org) [He et al., 2021] is a partially observable multi-agent domain, modeling a typical business organization that features a mix of individual competition with cooperation to improve the financial health of the organization. There are 5 (hidden) states corresponding to various levels of financial health, which map to 3 observations that an agent can receive, as shown in Fig. 1(left). An agent 0 has 3 action choices: self, group, and balance, the latter benefiting both the organization and the individual. Agent 0's reward is comprised of an individual (competitive) component, $R_0$, which depends on the agent's action ($a_0$),

and a group (cooperative) component, $R_G$, which depends on the joint action ($\boldsymbol{a}$). Let,

$$R_0^t \leftarrow R_0(s^t, a_0^t), \ \ R_G^t \leftarrow R_G(s^t, \boldsymbol{a}^t) \tag{1}$$

A vital feature of this domain is an additional *history-dependent* reward component, $R_{-1}$, that models a bonus payoff based on the organization's previous year performance, specifically a fraction $\phi$ of the previous reward,

$$R_{-1}^t = \phi(\sum_i R_i^{t-1} + R_G^{t-1}). \tag{2}$$

Joint actions are determined by the number of agents picking self compared to that picking group, and this affects the state transitions as shown in Fig. 1 (right). On the one hand, self action yields a higher individual reward to an agent than balance and group actions. But, it also damages the financial health of the organization if too many agents act in a self-interested manner. Moreover, group action improves the financial health at the expense of individual rewards. Thus, with the objective of optimizing $\mathbb{E}_{trajectories}\left[\sum_t \gamma^t(R_G^t + R_i^t + R_{-1}^t)\right]$, $i$ needs to balance greed with group welfare to optimize long-term payoff.

### 2.2 I-POMDP MODEL OF ORG

Multi-agent domains that involve frequent interactions among agents are typically modeled using the well-known interactive partially observable Markov decision process (I-POMDP) [Gmytrasiewicz and Doshi, 2005, Doshi, 2012], a framework that generalizes POMDPs [Kaelbling et al., 1998] to sequential decision-making in multi-agent environments. An I-POMDP for agent 0 with $N$ other agents in Org is defined as,

$$\text{I-POMDP}_0 = \langle IS_0, A, T_0, O_0, Z_0, \Omega_0, W_0, R_0 \rangle.$$

• $IS_0$ denotes the interactive state space, $IS_0 = \langle S_f, S_r \rangle \times \prod_{j=1}^N M_j$. This includes the physical (financial) state $S_f$, and the previous-step reward as an additional state feature $S_r$, as well as models of the other agent $M_j$, which may be intentional (ascribing beliefs, capabilities and preferences) or subintentional [Dennett, 1971]. Examples of the latter are probability distributions and finite state machines. In this paper, we ascribe subintentional models to the other agents, $m_j = \langle \pi_j, h_j \rangle$, $m_j \in M_j$, where $\pi_j$ is $j$'s policy and $h_j$ is its action-observation history.

• $A = A_0 \times \prod_{j=1}^N A_j$ is the set of joint actions of all agents. Let $\boldsymbol{a}_{-0}$ denote the joint actions of $N$ other agents, $\boldsymbol{a}_{-0} \in \prod_{j=1}^N A_j$.

• $T_0$ represents the transition function,

$$T_0(\langle s_f, s_r \rangle, a_0, \boldsymbol{a}_{-0}, \langle s_f', s_r' \rangle)$$
$$= \begin{cases} T(s_f, a_0, \boldsymbol{a}_{-0}, s_f'), & \text{if } s_r' = R(s_f, a_0, \boldsymbol{a}_{-0}) + \phi \cdot s_r \\ 0 & \text{otherwise} \end{cases}$$

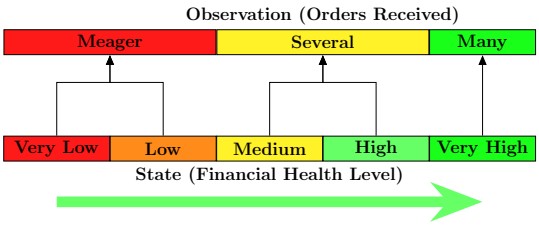
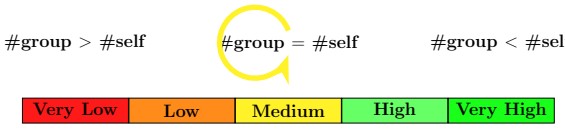

Figure 1: States, noisy observations, and transition dynamics of the Org domain.

The transition function is defined over the physical states and excludes the other agent's models as is standard for I-POMDPs. The function $T(s_f, a_0, \boldsymbol{a}_{-0}, s'_f)$ represents Org's Markovian transition dynamics.

• $O = O_f \times O_r$ is the set of *public* observations, where $O_f$ informs about the financial state and $O_r = S_r$.

• $Z_0$ is the observation function,

$$Z_i(a_0, \boldsymbol{a}_{-0}, \langle s_f, s_r \rangle, \langle s'_f, s'_r \rangle, \langle o'_f, o'_r \rangle)$$
$$= \begin{cases} Z(a_0, \boldsymbol{a}_{-0}, s_f, s'_f, o'_f), & \text{if } (s'_r = R(s_f, a_0, \boldsymbol{a}_{-0}) \\ & \quad + \phi \cdot s_r) \wedge (o'_r = s'_r) \\ 0 & \text{otherwise} \end{cases}$$

The observation function is defined over the physical state space only as a consequence of the model non-observability assumption in I-POMDPs. Here, $Z(a_0, \boldsymbol{a}_{-0}, s_f, s'_f, o'_f)$ models an agent's public observation function in Org.

• $\Omega_0$ is the set of agent 0's *private* observations.

• $W_0 : A \times \Omega_0 \to [0, 1]$ is the private observation function.

• $R_0$ defines the reward function for agent 0, $R_0(\langle s_f, s_r \rangle, a_0, \boldsymbol{a}_{-0}; \phi) = R(s_f, a_0, \boldsymbol{a}_{-0}) + \phi \cdot s_r$.

For a more detailed description of the Org domain and its I-POMDP model, see He et al. [2021].

The subject agent's belief is a distribution over the interactive state space, $b_0 \in \Delta(S_f \times \prod_{j=1}^N M_j)$. If the other agents' behaviors are known to be not correlated, this may be factorized as

$$b_0(s_f, m_1, \ldots, m_N) = b_0(s_f) b_0(m_1|s_f) \times \ldots \times b_0(m_N|s_f).$$

## 2.3 INTERACTIVE A2C FOR MIXED SETTINGS

Interactive advantage actor-critic (IA2C) [He et al., 2021] is a decentralized actor-critic method designed for egocentric RL in partially observable Markovian settings shared with other agents. In IA2C, each agent has its own critic and actor neural network, the former mapping individual observations to joint action values in terms of the agent's own reward function, $Q_0(o, a_0, \boldsymbol{a}_{-0})$ and the latter mapping individual observations to individual action probabilities, $\pi_{0,\boldsymbol{\theta}}(a_0|o)$, $\boldsymbol{\theta}$ is its set of parameters. Here, $o = \langle o_f, o_r \rangle$ is the public observation. IA2C estimates advantages as

$$A_0(o, a_0, \hat{\boldsymbol{a}}_{-0}) = avg[r + \gamma Q_0(o, a'_0, \hat{\boldsymbol{a}}'_{-0}) - Q_0(o, a_0, \hat{\boldsymbol{a}}_{-0})]$$

while the actor's gradient is estimated as

$$avg[\nabla_{\boldsymbol{\theta}} \log \pi_{0,\boldsymbol{\theta}}(a_0|o) \, A_0(o, a_0, \hat{\boldsymbol{a}}_{-0})]$$

where $r, o$ and $a'_0$ are samples, $\hat{\boldsymbol{a}}_{-0}$ and $\hat{\boldsymbol{a}}'_{-0}$ are *predicted* actions, and the $avg$ is taken over sampled trajectories. In contrast with previous multi-agent deep RL algorithms, IA2C does not require access to other agents' actions and/or gradients. Rather, agents maintain belief distributions over other agents' possible models. Let $\boldsymbol{a}_{-0} = a_1, \ldots, a_N$. Given the agent's prior belief $b_0$, action $a_0$, as well as its public and private observations $o'_0, \omega'_0$, the agent updates its belief over agent $j$'s model for $m'_j = \langle \pi'_j, h'_j \rangle$ as

$$b'_0(m'_j|b_0, a_0, o', \omega'_0) \propto \sum_{\boldsymbol{a}_{-0}} \Bigg( \prod_{k=1, k \neq j}^N \sum_{m_k \in M_k} b_0(m_k)$$
$$Pr(a_k|m_k) \sum_{m_j \in M_j} b_0(m_j) \, Pr(a_j|m_j) \, \delta_K(\pi'_j, \pi_j)$$
$$\delta_K(APPEND(h_j, \langle a_j, o' \rangle), h'_j) \Bigg) W_0(a_0, \boldsymbol{a}_{-0}, \omega'_0) \quad (3)$$

where public and private observations are noisy observations of states and other agents' actions. $m_j$ denotes agent $j$'s model, $\pi_j$ is $j$'s policy and $h_j$ is its action-observation history. $W_0$ is the private observation function that maps joint actions to private observations. $\delta_K$ is the Kronecker delta function and APPEND returns a string with the second argument appended to its first. The belief update is performed by a belief filter integrated into the critic network.

## 3 MANY-AGENT RL UNDER PARTIAL OBSERVABILITY

The primary challenge in scaling multi-agent RL to many agents is the exponential growth of the joint action space. However, if the population is *homogeneous* [1] in that all the agents have the same action space ($A_0 = A_1 = \ldots = A_N$) and the domain exhibits the *action anonymity* property, which means that both the dynamics and the rewards depend

---

[1]We define homogeneity broadly to compute configurations or mean actions from joint actions. Although agents may have different models, configuration or mean action can be obtained as long as all agents have the same action space.

on the count distribution of actions in the population, while not needing the agents' identities, then we may scale.

## 3.1 IA2C USING ACTION CONFIGURATIONS

Under the conditions of population homogeneity and action anonymity, we present an alternate way to model large agent populations, which has been effective in scaling decision-theoretic planning.

### 3.1.1 Action Configurations

We begin by defining the concept of a configuration and characterize its properties.

**Definition 1** (Configuration). *Define a configuration denoted by $\mathcal{C}$ as a vector of counts of the distinct actions performed by the $N$ agents at a time step, $\mathcal{C}^{\boldsymbol{a}} = \langle \#a^1, \#a^2, \ldots, \#a^{|A|} \rangle$, where $\#a^1$ denotes the count of an action $a^1$ in the joint action $\boldsymbol{a}$ and $\#a^1 \leq N$. Denote by $\mathcal{C}$ the finite set of all configurations.*

**Definition 2** (Projection). *Define a projection function $\delta$ as a mapping $\delta : A \to \mathcal{C}$, which maps a joint action to its corresponding configuration.*

For example, $\delta$ projects the joint action $\boldsymbol{a} = \langle$ self, self, group, group $\rangle$ in the Org domain to the configuration vector, $\mathcal{C}^{\boldsymbol{a}} = \langle 2, 2, 0 \rangle$, where the vector components give the counts of actions self, balance, and group, respectively. This is analogous to the *mean action* $\bar{a} = \langle \frac{1}{2}, \frac{1}{2}, 0 \rangle$ [Yang et al., 2018], obtained by dividing the action configuration by the number of agents.

Observe that $\delta$ is a *many-one mapping* as multiple distinct joint actions, which are permutations of each other, yield the same configuration vector. In other words, for any $s$, $a_0$, $s'$, $\boldsymbol{a}_{-0}$, and a permutation of $\boldsymbol{a}_{-0}$ denoted as $\dot{\boldsymbol{a}}_{-0}$, we have:

$$T_0(s, a_0, \boldsymbol{a}_{-0}, s') = T_0(s, a_0, \dot{\boldsymbol{a}}_{-0}, s') = T_0(s, a_0, \mathcal{C}^{\boldsymbol{a}_{-0}}, s'),$$
$$Z_0(a_0, \boldsymbol{a}_{-0}, s, s', o') = Z_0(a_0, \dot{\boldsymbol{a}}_{-0}, s, s', o') = Z_0(a_0, \mathcal{C}^{\boldsymbol{a}_{-0}}, s, s', o'),$$
$$W_0(a_0, \boldsymbol{a}_{-0}, \omega_0') = W_0(a_0, \dot{\boldsymbol{a}}_{-0}, \omega_0') = W_0(a_0, \mathcal{C}^{\boldsymbol{a}_{-0}}, \omega_0'),$$
$$R_0(s, a_0, \boldsymbol{a}_{-0}) = R_0(s, a_0, \dot{\boldsymbol{a}}_{-0}) = R_0(s, a_0, \mathcal{C}^{\boldsymbol{a}_{-0}})$$

where $\delta(\boldsymbol{a}_{-0}) = \delta(\dot{\boldsymbol{a}}_{-0}) = \mathcal{C}^{\boldsymbol{a}_{-0}}$. As such, we may not recover the original joint action back from the configuration – a direct consequence of the action anonymity property. The above equivalences naturally lead to the following property of the Q-function:

$$Q_0(o, a_0, \boldsymbol{a}_{-0}) = Q_0(o, a_0, \dot{\boldsymbol{a}}_{-0}) = Q_0(o, a_0, \mathcal{C}^{\boldsymbol{a}_{-0}}).$$

Subsequently, the advantage function $A_0(o, a_0, \boldsymbol{a}_{-0})$ is also rewritten with the projection to the configuration. *A key advantage of using configurations is that the space of vectors of action counts is polynomial in the number of agents in comparison to the exponential growth of the joint action space as the number of agents grows.*

Notice that action anonymity is a domain feature, independent of the modeling approach. Several many-agent domains naturally exhibit the anonymity-driven joint action permutation invariance property. In addition to Org, consider an intelligent traffic control system that adjusts the traffic light duration according to the number of vehicles entering and leaving a congested intersection. The identities of the vehicles would not be required here. We adapt the IA2C of Sec. 2.3 to replace joint actions with action configurations to enable many-agent RL in partially observable settings, and label this new method as IA2C$^{++}$. IA2C's belief filter is modified and a new dynamic programming module is prepended to the belief filter in the critic.

### 3.1.2 Modeling Agent Populations

Equation 3 in Section 2.3 gives the update of agent 0's belief over *one* other agent's possible models $M_j$, and this is performed for each other agent $j = \{1, 2, \ldots, N\}$ – growing linearly in $N$. Joint action in the private observation function $W_0$ is now replaced by action configuration, as introduced previously. But, this also necessitates an additional term in the equation as we show below.

$$b_0'(m_j'|b_0, a_0, o', \omega_0') \propto \sum_{m_j \in M_j} b_0(m_j) \sum_{a_j} Pr(a_j|m_j)$$
$$\times \sum_{\mathcal{C} \in \mathcal{C}^{\boldsymbol{a}_{-0}}} Pr(\mathcal{C}|b_0(M_1), b_0(M_2), \ldots, b_0(M_N)) \times$$
$$W_0(a_0, \mathcal{C}, \omega_0') \delta_K(\pi_j, \pi_j') \, \delta_K(APPEND(h_j, \langle a_j, o' \rangle), h_j').$$
$$(4)$$

Here, $Pr(\mathcal{C}|b_0(M_1), b_0(M_2), \ldots, b_0(M_N))$ is the probability of a configuration in the distribution over the set of configurations $\mathcal{C}^{\boldsymbol{a}_{-0}}$. The distribution is obtained from agent 0's factored beliefs over the models of each other agent using a known dynamic programming procedure [Jiang et al., 2011] that is outlined in Algorithm 1 presented in the supplementary material. The algorithm takes as input just $N$ beliefs each of size $|M_j|$ compared to a single large belief of exponential size $|M_j|^N$, which is a benefit of the belief factorization shown in Section 2.3.

By updating beliefs using Eq. 4, each agent maintains belief distributions over each other agents' models. However, we still need to have either true models of other agents or a pre-defined model set to predict other agents' actions. In the case that we do not have enough information about the domain to manually construct a meaningful pre-defined set of models for Eq. 4, the accuracy of action prediction may be insufficient for the learning algorithm to converge. In such scenarios, the Dirichlet-multinomial distribution can be used for modeling the agent population. It has been widely applied for modeling categorical variables and has

found its way into machine learning areas such as natural language processing [Blei et al., 2002].

Suppose the action space for the homogeneous agent population is $\{a^1, a^2, \ldots, a^{|A|}\}$, and for each agent $i$, the probability of picking action $a^n$ is $\theta_n$. Let the distribution $\boldsymbol{\theta} = (\theta_1, \theta_2, \ldots, \theta_{|A|})$. $\boldsymbol{\theta}$ has a Dirichlet-multinomial prior distribution with parameter $\boldsymbol{\alpha}$ if

$$Pr(\boldsymbol{\theta}|\boldsymbol{\alpha}) = \frac{\Gamma(\sum_n \alpha_n)}{\Pi_n \Gamma(\alpha_n)} \Pi_n \theta_n^{\alpha_n - 1} \qquad (5)$$

where $\alpha_n > 0$ for all $n$, $\boldsymbol{\alpha} = (\alpha_1, \alpha_2, \ldots, \alpha_N)$, and $\sum_n \theta_n = 1$. We can conveniently write this as $\boldsymbol{\theta} \sim Dirichlet(\boldsymbol{\alpha})$. Then, probability of an action configuration $\mathcal{C}$ is expressed as:

$$Pr(\mathcal{C}|\boldsymbol{\theta}) = \Pi_{n=1}^{|A|} \theta_n^{\#a^n}.$$

On performing action $a_0$ and receiving a (noisy) private observation of $\omega_0' = (\#a^1, \#a^2, \ldots, \#a^{|A|})'$, the subject agent's Dirichlet-multinomial distribution can be updated by noting that the posterior $Pr(\boldsymbol{\theta}|\omega_0', a_0)$ is proportional to:

$$\propto Pr(\omega_0'|\boldsymbol{\theta}, a_0) \, Pr(\boldsymbol{\theta}|a_0) = Pr(\omega_0'|\boldsymbol{\theta}, a_0) \, Dirichlet(\boldsymbol{\alpha})$$

$$= \sum_{\mathcal{C}} Pr(\mathcal{C}, \omega_0'|a_0, \boldsymbol{\theta}) \, Dirichlet(\boldsymbol{\alpha})$$

$$= \sum_{\mathcal{C}} Pr(\omega_0'|a_0, \mathcal{C}, \boldsymbol{\theta}) Pr(\mathcal{C}|a_0, \boldsymbol{\theta}) \, Dirichlet(\boldsymbol{\alpha})$$

$$= \sum_{\mathcal{C}} W_0(a_0, \mathcal{C}, \omega_0') \, Pr(\mathcal{C}|\boldsymbol{\theta}) \, Dirichlet(\boldsymbol{\alpha}) \qquad (6)$$

$$\propto \sum_{\mathcal{C}} W_0(a_0, \mathcal{C}, \omega_0') \left( \Pi_{n=1}^{|A|} \theta_n^{\#a^n} \right) \left( \Pi_{n=1}^{|A|} \theta_n^{\alpha_n - 1} \right)$$

$$= \sum_{\mathcal{C}} W_0(a_0, \mathcal{C}, \omega_0') \Pi_{n=1}^{|A|} \theta_n^{\#a^n + \alpha_n - 1}$$

$$= \sum_{\mathcal{C}} W_0(a_0, \mathcal{C}, \omega_0') \, Dirichlet(\boldsymbol{\alpha} + \mathcal{C}) \qquad (7)$$

$$\approx Dirichlet(\boldsymbol{\alpha} + \mathcal{C}'). \qquad (8)$$

In (6), we utilize the fact that $\omega_0'$ is conditionally independent of $\boldsymbol{\theta}$ given $a_0, \mathcal{C}$, and therefore $Pr(\omega_0'|a_0, \mathcal{C}, \boldsymbol{\theta})$ is in fact $W_0(a_0, \mathcal{C}, \omega_0')$. We also make a simplifying modeling assumption in (6) that the true action counts of other agents, $\mathcal{C}$, is conditionally independent of $a_0$ given $\boldsymbol{\theta}$. For the sake of tractability, we approximate the last step as a single component, $Dirichlet(\boldsymbol{\alpha} + \mathcal{C}')$. We investigate different alternative ways of calculating $\mathcal{C}'$ later in this subsection. Equation 8 allows each agent to maintain and update its own Dirichlet-multinomial distribution over the agent population; the predefined model set and individual belief updates introduced earlier in this section are no longer required.

In this paper, we consider a common model of noise where an agent observes another agent's true action with probability $1 - \delta$ (for a small $\delta$), but observes a different action with

probability mass $\delta$ distributed uniformly over the remaining $|A| - 1$ actions. Under this model, surprisingly, the cumulative effect of the noise over $N$ agents may decrease with $N$, instead of increasing (e.g., with multivariate Gaussian noise). The proof of the following proposition is included in the supplementary material.

**Proposition 1.** *The probability of error (which occurs when an observed configuration $\omega_0'$ is different from the true configuration $\mathcal{C}$) is a decreasing function of $N$ if $N > \frac{|A|}{\log(1/1-\delta)}$, where $\delta$ is as defined previously.*

Next, we consider the following three methods for calculating $\mathcal{C}'$ in Eq. 8.

**Naive:** In this method, we ignore $W_0$ and use the observed configuration directly to set $\mathcal{C}' \leftarrow \omega_0'$, despite $\omega_0'$ being noisy.

**Rectified:** In this method, we proportionately undo the noise introduced by the $\delta$-noise model. Suppose, $\mathcal{C}' = (\#a^{1'}, \ldots, \#a^{|A|'})$. We expect $(1 - \delta)$ fraction of $\#a^{i'}$ to contribute to $\omega_0'[i]$, while the contribution of $\#a^{j'}$ to $\omega_0'[i]$ is expected to be $\frac{\delta}{|A|-1}, \forall j \neq i$. As $\omega_0'$ is given, this yields a system of linear equations for $i = 1 \ldots, |A|$,

$$\omega_0'[i] = (1 - \delta)\#a^{i'} + \sum_{j \neq i} \frac{\delta}{|A| - 1} \#a^{j'} \qquad (9)$$

solving which we can recover the estimated true configuration, $\mathcal{C}'$. Furthermore, as these counts get accumulated by $\boldsymbol{\alpha} \leftarrow \boldsymbol{\alpha} + \mathcal{C}'$, increasing the samples on the left hand side of the above equations, we expect the solutions to become increasingly accurate.

**Optimized:** In this method, we set $\mathcal{C}' \leftarrow \arg\max_{\mathcal{C}} W_0(a_0, \mathcal{C}, \omega_0') Dirichlet(\boldsymbol{\alpha} + \mathcal{C})$. This optimization reduces the error between the true posterior (Eq. 7) and its approximation (Eq. 8) while keeping the posterior computation scalable. It yields an iterative procedure for updating $\mathcal{C}$ based on the gradient of the (log of the) above objective as shown below.

$$\Delta \#a^k = \mathcal{P} \left( \frac{\partial \log W_0}{\partial \#a^k} + \psi(\alpha_0 + N) - \psi(\alpha_k + \#a^k) + \right.$$
$$\left. \log \theta_k \right)$$

where $\alpha_0 = \sum_k \alpha_k$, $\psi$ is the digamma function, $\psi(x) = \frac{d \log \Gamma(x)}{dx}$, and $\mathcal{P}$ ensures the projection of $\mathcal{C}$ onto the plane $\sum_i \#a^i = N$. This method is sufficiently general to apply to different noise models. For instance, if the observation noise is multivariate Gaussian, $\mathcal{N}(\mathcal{C}, \Sigma)$, then $\frac{\partial \log W_0}{\partial \mathcal{C}} = \Sigma^{-1}(\omega_0' - \mathcal{C})$. For our $\delta$-noise model, we approximate $\frac{\partial \log W_0}{\partial \mathcal{C}} \approx (\omega_0' - \omega_{\mathcal{C}})$, where $\omega_{\mathcal{C}}$ is the left hand side from Eq. 9 obtained by plugging in $\mathcal{C}$ into its right hand side.

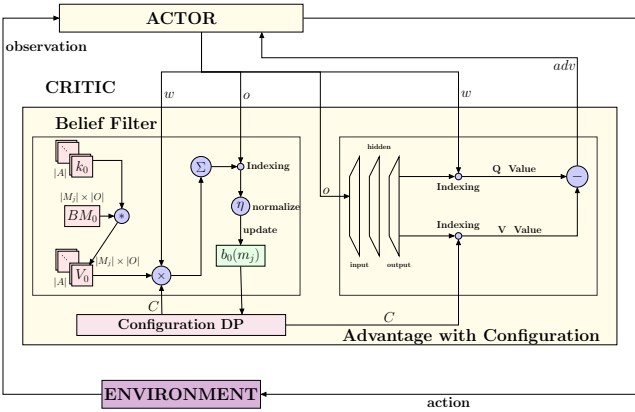

Figure 2: The belief filter in the *critic* utilizes the distribution over configurations from dynamic programming to update the agent's belief over models. Predicted actions are projected to their corresponding configuration and used in obtaining the advantage. The belief filter module is subsequently replaced with the Dirichlet-multinomial update.

## 3.2 MANY-AGENT INTERACTIVE A2C

We illustrate the many-agent IA2C with belief update (labeled as IA2C$^{++}$BU) architecture in the schematic of Fig. 2. It consists of two main components: the actor and the critic. The expression in (10) below gives the actor's revised gradient, which is updated as the subject agent 0 interacts with the environment:

$$avg\left[\nabla_{\boldsymbol{\theta}} \log \pi_{0,\boldsymbol{\theta}}(a_0|o) A_0(o, a_0, \mathcal{C}^{\boldsymbol{a}_{-o}})\right]. \quad (10)$$

Notice that the actions of the other agents typically appearing in the advantage function are now replaced with its projected configuration $\mathcal{C}^{\boldsymbol{a}_{-o}}(= \delta(\boldsymbol{a}_{-o}))$. This new advantage function is computed by the critic as:

$$A_0(o, a_0, \mathcal{C}^{\boldsymbol{a}_{-o}}) = avg[r + \gamma Q_0(o', a'_0, \mathcal{C}^{\boldsymbol{a}'_{-o}}) - Q_0(o, a_0, \mathcal{C}^{\boldsymbol{a}_{-o}})]$$

where $r$, $o'$, and $a'_0$ are samples, $\boldsymbol{a}_{-0}$ and $\boldsymbol{a}'_{-0}$ are the *predicted* most-likely joint actions of the other agents for the current and next step, respectively, replaced by their corresponding configurations, and $avg$ is taken over the sampled trajectories. An agent $j$'s predicted action for the next time step is obtained by first sampling its model from the updated $b'_0(m'_j)$, where the update occurs as per Eq. 4. The sampled model yields an action distribution using which the action is sampled. This procedure is performed for each other agent and the corresponding configuration is obtained.

Thus, the actor network forwards the public and private observations from the environment to the belief filter in the critic. The belief filter first runs the dynamic programming procedure, uses the output distribution over configurations to then update agent 0's belief over other agents' models, and predicts their actions. The projection operator yields the corresponding current and next time-step configurations, all

of which is sent to the critic neural net for gradient-based updating and then to the advantage module for computing the advantage function. The latter is sent back to the actor component for its gradient update.

The IA2C with Dirichlet-multinomial distribution (labeled as IA2C$^{++}$DM) replaces the belief update module in IA2C$^{++}$BU with a Dirichlet-multinomial distribution. The network architecture is similar to IA2C$^{++}$BU except that the belief filter and the dynamic programming are replaced with the Dirichlet-multinomial distribution and its update. The Dirichlet is updated by *rectified* private observations at each time step, as described in Sec. 3.1.2. Action configurations are sampled from the updated distributions for gradient updates in both critic and actor networks. Unlike the belief update module in IA2C$^{++}$BU, the update filter in IA2C$^{++}$DM can be easily added to most existing actor-critic methods without requiring a pre-defined model set.

For both IA2C$^{++}$BU and IA2C$^{++}$DM, we implement the actor neural network with one input layer for the observations, two hidden layers, one with $tanh$ and the other with ReLU activation, followed by the output layer. The critic network consists of one input layer for the observations, one hidden layer with tanh activation followed by the output layer. All layers are fully connected to the next layer.

## 4 EXPERIMENTS

Our first experiment domain is the Org domain [He et al., 2021], which is a partially observable cooperative-competitive domain where the number of agents can be easily scaled up arbitrarily without stretching the domain semantics. We select five organization structures as shown in Fig. 3, which differ in the number of neighborhoods and the number of agents in each neighborhood. For example, in the fully connected structure, all agents share one single neighborhood; on the contrary, each agent forms a neighborhood with its left and right neighbors in the circle structure.

Our second experiment domain is a modified version of the battlefield in the MAgent environment [Zheng et al., 2018]. The battle field in Fig. 4a is separated into four sub-fields. Five agents each from red and blue teams are deployed in each sub-field. Each agent may choose to move to an adjacent grid from its current location or attack a nearby opponent. An agent can observe the entire field. The reward function consists of two parts: individual rewards and group rewards. An agent receives -1 for moving or attacking and +10 for eliminating an opponent. An agent receives a group reward of +2 if its team has more alive members than the opponent team in each time step. Otherwise, the agent receives -1 as the group reward. Our implementation of IA2C$^{++}$ is available in Python on GitHub at https://github.com/khextendss/mIA2C.

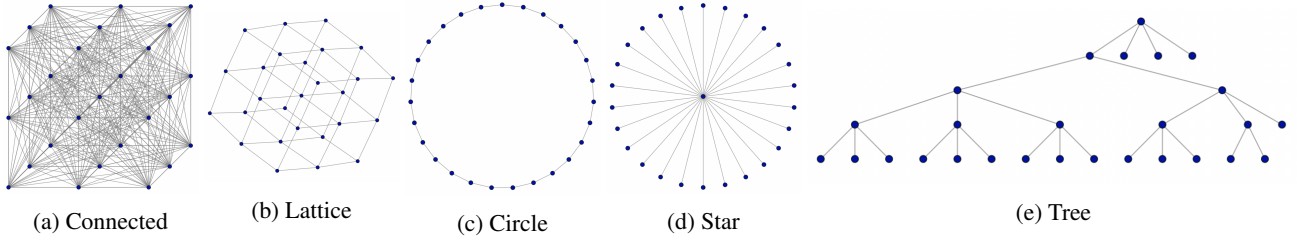

(a) Connected  (b) Lattice  (c) Circle  (d) Star  (e) Tree

Figure 3: Various common interaction topologies for the Org domain with 27 agents.

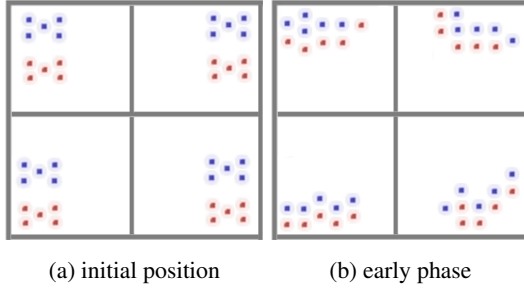

(a) initial position  (b) early phase

Figure 4: The battlefield is separated into four sub-fields. ($a$) Agents from blue and red teams are initially deployed in the corners. ($b$) Agents start battling with opponents.

### 4.1 COMPARISON OF METHODS FOR UPDATING DIRICHLET-MULTINOMIAL

We sample the agents' actions from a fixed mixture distribution, and add $\delta$-noise to them to generate $\omega'_0$. Then, the three methods outlined in Sec. 3.1.2 yield their estimates of $\mathcal{C}'$, to update the Dirichlet parameters as $\alpha \leftarrow \alpha + \mathcal{C}'$. This is continued for 20 steps in a trial. We compare the learned distributions $Dirichlet(\alpha + \mathcal{C}')$ with the true mixture distribution, and obtain the prediction accuracy measure for each of the 20 steps across 5 independent trials. The results are shown in Figs. 5a (a), 5b (b) for 20%- and 50%-noise.

These formative experiments with the three methods show that both Rectified and Optimized $\mathcal{C}'$ have similar predictive accuracy, while the Naive estimate is significantly less accurate. As the Rectified method is much faster than Optimized, we use the former in our subsequent experiments.

### 4.2 IMPACT OF ORG TOPOLOGIES

We instantiate IA2C with action configuration. Agents receive group rewards from the neighborhood. We explore the performance of agents learning using IA2C in the five topologies of agent connectivity in Org as shown in Fig. 3(a-e). Our aim is to understand the impact of the differing neighborhoods in these topologies on the overall learned reward for the organization. Figure 6 shows the cumulative rewards, summed over all agents in the organization, obtained from executing the converged policies learned by

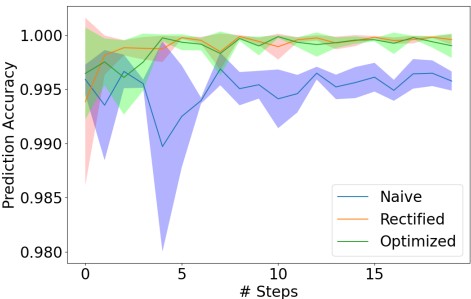

(a) Prediction accuracy over 20 steps when $\delta = 0.2$.

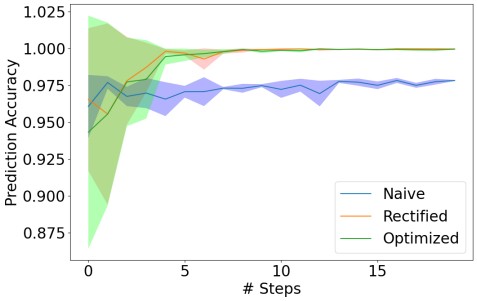

(b) Prediction accuracy over 20 steps when $\delta = 0.5$.

Figure 5: Prediction accuracy comparison between naive, rectified, and optimized methods for obtaining $\mathcal{C}'$.

IA2C. We evaluate this metric for an increasing number of agents. We observe that the cumulative rewards in the fully-connected structure are higher than the tree, lattice, circle, and star structures. Moreover, IA2C converges to policies with higher cumulative rewards in lattice and circle structures than the star and tree topologies. This can be explained by noting that the star and tree structures contain many neighborhoods of only two agents (leaf nodes), which makes it challenging for those agents to coordinate across these small neighborhoods without knowing the actions of agents outside their neighborhoods. As such the fully-connected structure shows the best performance where everybody knows what everybody is doing, but, of course, the computational complexity is greater causing IA2C to converge relatively slowly.

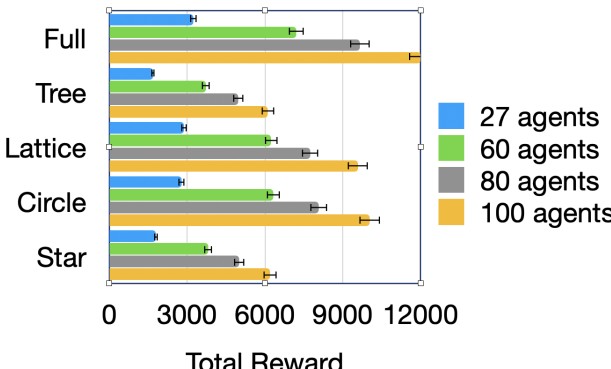

Figure 6: Averaged cumulative rewards comparison between various Org topologies for 27, 60, 80, and 100 agents.

### 4.3 MODELING OTHERS

Next, we compare the performance of the previous belief update (IA2C$^{++}$BU) to the Dirichlet-multinomial distribution (IA2C$^{++}$DM) in terms of prediction accuracy and run time. We measure the accuracy by the KL-divergence between predicted action distribution and the true action distribution. Figure 7a shows KL-divergence of both methods for the different number of agents under various private observation noises after 1,000 updates. The belief update method has lower divergence than the Dirichlet-multinomial distribution when $\delta$-noise is less than 0.2 for 10 and 100 agents. Due to the small sample size, the Dirichlet-multinomial distribution does not perform well for 10 gents. It achieves comparable divergence for 100 agents and has lower divergence than belief update for 1,000 agents. It also shows the ability to maintain accurate predictions in more noisy environments. Figure 7b shows a run time comparison of the two methods with $\delta = 0.4$. The run time of the Dirichlet-multinomial distribution increases linearly with respect to the number of agents while having a small slope. The run time of belief update increases significantly faster since the belief update process needs to be done for every single agent and the extra computation required by dynamic programming for configuration distribution.

As a result, the belief update is better suited for modeling small agent population with less than 0.3 private observation noise; the Dirichlet-multinomial distribution works well with large agent populations and noisier environments.

### 4.4 PERFORMANCE COMPARISON

We compare IA2C$^{++}$BU and IA2C$^{++}$DM with the mean-field actor-critic [Yang et al., 2018] (MF-AC) and QMIX [Rashid et al., 2018] on the tree, star, and fully connected Org and MAgent battlefield with 100 agents. QMIX uses the sum of mean rewards of all agents as the team reward. We limit the run time to 20 hours and set the private

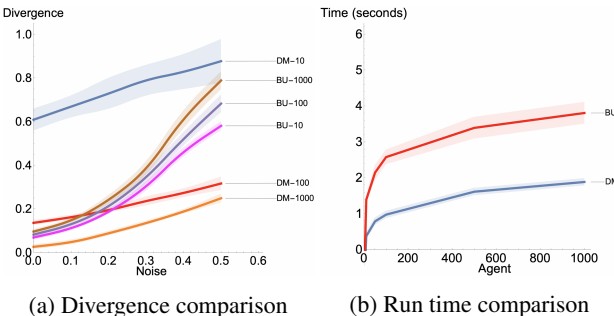

(a) Divergence comparison  (b) Run time comparison

Figure 7: (a) KL-divergence of the belief update method and Dirichlet-multinomial distribution method for 10, 100, and 1,000 agents. The private noise probability denotes the probability of receiving a noised private observation. (b) Run time of belief update and Dirichlet distribution update. All experiments are conducted on a standard Linux PC with Intel i7 processor (8 cores, 3.6 GHz) and 64 GB memory.

observation noise at 0.2. [2]

Figure 8 shows cumulative rewards and win rates of policies learned by all methods in Org and MAgent battlefield. IA2C$^{++}$DM converges to the optimal policy within the shortest time. The large population size causes the belief update process in IA2C$^{++}$BU to slow down the learning. However, IA2C$^{++}$DM benefits from the large agent population. The Dirichlet multinomial model accurately and efficiently predicts the configuration of the agent population. Due to the noised private observations, MF-AC and QMIX do not converge to optimal policy within the given time limit.

### 5 RELATED WORK

Mean-field Q-learning (MF-Q) [Yang et al., 2018] is a Q-learning method that scales to many agents. If the agents are indistinguishable and independent from each other, they are substituted by a single virtual agent which performs the mean action. The Q-value of the state and joint action is approximated by the Q-value of the state and the mean action. A mean-field actor-critic (MF-AC) is also presented, but experiments show that MF-AC rarely improves on MF-Q. Par-

---

[2] We use time in hours as the measure on the x-axes rather than episodes because the methods utilize different ways of scaling and tackling partial observability. Number of episodes may conceal critical information about algorithm efficiency. For example, IA2C$^{++}$BU and IA2C$^{++}$DM spend time on modeling the agent population via belief update and Dirichlet distribution update, respectively. On the other hand, QMIX and MF-AC infer other agents' actions by MLE, which is faster. However, the noisy observations makes them require more episodes to converge. As such, measures of elapsed (clock) time accounts for all such considerations. Nevertheless, we show cumulative rewards with respect to episodes as well in the supplementary material.

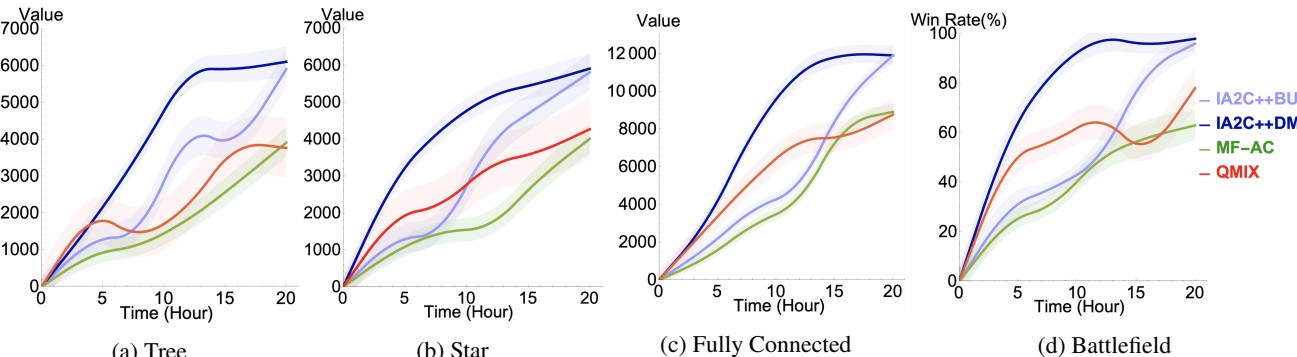

Figure 8: Cumulative reward of learned policies in (*a*) tree structure, (*b*) star structure, and (*c*) fully connected structure. (*d*) Win rate against pre-trained agents in the MAgent battlefield domain. All experiments are conducted on a standard Linux PC with Intel i7 processor (8 cores, 3.6 GHz) and 64 GB memory.

tially observable mean-field Q-learning (POMF-Q) Ganapathi Subramanian et al. [2021] extends MF-Q to partially observable settings. Each agent maintains a Dirichlet distribution for current population and the mean action is sampled from the Dirichlet. Although POMF-Q adopts a similar way to model agent populations, other agents' actions are perfectly observed (i.e., noiseless private observations) and the algorithm is specifically designed for spatial domains. In contrast, IA2C$^{++}$DM is not limited to spatial domains and permits noisy private observations, which is realistic but makes the Dirichlet-multinomial update much less straightforward.

In contrast to our use of action configurations, Verma, Varakantham, and Lau [Verma et al., 2019] utilize state configurations under anonymity in the form of counts of agents located in various zones (the agents are taxis). Other agents' actions in the transition and reward functions are now replaced by state configurations without loss of information in the taxi domain. Algorithms that extend deep Q-networks and A2C to these settings are presented. However, the state is perfectly observed obviating the need for distributions over configurations, and the experiments were mostly limited to 20 agents operating in about 100 zones.

A recent scalable MARL technique is QMIX [Rashid et al., 2018] which utilizes neural networks to estimate the joint action value function as a nonlinear combination of individual agent value functions. At the same time, QMIX enforces a monotonic relationship between a centralized Q-value and individual Q-values, which allows the joint-action value maximization in off-policy learning to be tractable. Nevertheless, QMIX uses an unweighted mixing projection in cooperative settings, which can lead to suboptimal policies. Weighted-QMIX [Rashid et al., 2020] overcomes this issue by infusing a weighting function into the projection from the joint action to the mixing network. However, as we showed in the previous section, the method may not be robust to noisy observations of others' actions.

## 6 CONCLUDING REMARKS

We presented new scalable actor-critic MARL algorithms built on the property of action anonymity that scales polynomially with the number of agents. Owing to a quadratic-time Dirichlet-multinomial distribution approach for modeling agent populations under partial observability, our method is able to accurately and efficiently predict action configurations for large agent populations. When compared to recent many-agent MARL methods that assume other agents' policies or actions are observed accurately, our methods show superior performance in terms of the quality of learned policies and convergence speed.

The noise-based exploration employed in recent MARL algorithms can be slow. In this regard, Liu, Jain, Yeh, and Schwing [Liu et al., 2021] propose cooperative multi-agent exploration (CMAE), which introduces a bottom-up exploration scheme that projects the high-dimensional state space to low-dimensional spaces and gradually explores from low- to high-dimensions. Empirical results show that CMAE increases exploration efficiency, particularly in sparse-reward environments. Integrating CMAE into IA2C's explorations could be an interesting avenue for future work to gain faster convergence.

### Acknowledgements

This work was supported in part by a grant from NSF #IIS-1910037 (to PD) and UGA's RIAS program for graduate students. We thank Nitay Alon for his technical feedback during a seminar presentation at HUJI and the anonymous reviewers for their valuable suggestions.

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
