# OpenReview forum: "Reinforcement Learning in Many-Agent Settings Under Partial Observability"
_auai.org/UAI/2022/Conference — UAI 2022 Poster_

### Official Review · Reviewer_J2n4 · 2022-04-01

**Q2(1) Originality/Novelty:** 2
**Q2(2) Significance/Impact:** 2
**Q2(3) Correctness/Technical Quality:** 2
**Q2(6) Clarity Of Writing:** 2
**Q6 Overall Score:** 6
**Q8 Confidence In Your Score:** 3

**Q1 Summary And Contributions:**

This paper studies the planning in factored partially observed Markov decision process (POMDP) where there exist multiple POMDP chains; each chain represents an agent. Every agent could make decisions based on its own observations and the observed trajectories of other agents. The goal is to learn optimal policies determining actions of all agents so that the overall performance of the system is maximized. This line of work is also referred to as multi-agent enforcement learning (MARL).

**Q2 Assessment Of The Paper:**

More detailed information regarding each of these aspects is given below:

**Q2(4) Quality Of Experiments (Optional):**

3: Good: The experimental evaluation is adequate, and the results convincingly support the main claims.

**Q2(5) Reproducibility:**

1: Poor: Key details (e.g., proof sketches, experimental setup) are incomplete/unclear, or key resources (e.g., proofs, code, data) are unavailable.

**Q3 Main Strengths:**

A continuing limitation of much of the work in MARL is the curse of dimensionality when it comes to representations based on joint actions, which grow exponentially with the number of agents. This paper attempts to address this challenge by exploiting the property of action anonymity. More specifically, the authors assume that reward function and transition probabilities are invariant over joint action permutations. By modeling this invariance enables the planning complexity to drop from being exponential in the number of agents to polynomial thereby facilitating multiagent planning for thousands of agents.

**Q4 Main Weakness:**

The presentation of this paper could be improved. More specifically, the description of the key insight, the invariance over action permutations, in Section 3.1.1 is somewhat confusing. The authors seem to imply that these are natural properties derived from IPOMDP. However, it seems to me that the invariance properties presented in Section 3.1.1 do not generally hold true in IPOMDPs. Are these additional assumptions imposed on the environment? If so, how practical are these assumptions? I would appreciate it if the authors could elaborate on this.

**Q5 Detailed Comments To The Authors:**

Please see Q4 for details.

**Q7 Justification For Your Score:**

This paper studies an interesting problem in planning in MARL settings. However, the presentation of the key insight, the invariance of reward functions and transition probabilities over action permutation, is somewhat confusing. It is unclear if there are inherent properties of the model. If these are indeed additional assumptions, I am afraid that they might be too restrictive to be practical. Consequently, the significance of the proposed method might be limited.

**Q9 Complying With Reviewing Instructions:**

1: Yes.

---

### Official Review · Reviewer_yYTP · 2022-04-13

**Q2(1) Originality/Novelty:** 2
**Q2(2) Significance/Impact:** 2
**Q2(3) Correctness/Technical Quality:** 3
**Q2(6) Clarity Of Writing:** 3
**Q6 Overall Score:** 6
**Q8 Confidence In Your Score:** 3

**Q1 Summary And Contributions:**

This paper extends IA2C to environments where the local transition and reward functions don't depend on the other agents’ individual actions but on the number of agents that selected each action (called configurations). They leverage the additional structure of the problem to derive an alternative to the belief update of the original algorithm. The belief update is then replaced with a Dirichlet-multinomial dist. leading to a mixture of Dirichlets in the posterior, approximated by 1 Dirichlet.

**Q2 Assessment Of The Paper:**

More detailed information regarding each of these aspects is given below:

**Q2(4) Quality Of Experiments (Optional):**

3: Good: The experimental evaluation is adequate, and the results convincingly support the main claims.

**Q2(5) Reproducibility:**

3: Good: Key resources (e.g., proofs, code, data) are available and key details (e.g., proofs, experimental setup) are sufficiently well-described for competent researchers to confidently reproduce the main results.

**Q3 Main Strengths:**

The paper introduces two algorithms that leverage their environments/setting structure.

The second algorithm is an efficient approximation of the first, which improves speed without degrading the performance and is more resistant to noise.

They provide a detailed analysis of which algorithm to use in different settings based on their experiments.


**Q4 Main Weakness:**

There is no comparison with the original algorithm IA2C.

The x-axis of the experimental plot is the time in hours. This is the first time I have seen this, and I do not think it is a good metric as it highly depends on how the algorithms are coded. While I understand that the authors want to show the time efficiency of their new method, it doesn’t allow a fair comparison. In my opinion, the number of timesteps/interactions would make more sense.

QMIX is a multi-agent RL algorithm for Dec-POMDPs (all the agents receive the same team reward), while the environments used are both Partially Observable Markov Games. The paper does not explain how they derive a team reward, or how the training is done. Could you please clarify this?

The original algorithm, IA2C, compares against MADDPG, an algorithm for POMGs. Why is it not the case here?


**Q5 Detailed Comments To The Authors:**

I would like to know possible applications of this algorithm as the setting of I-POMDP with action anonymity seems rather restrictive. Could you please provide a real-world example of such a setting? What are the limitations of this setting compared to general I-POMDPs, Dec-POMPDs and POMGs?

Figure 2 is rather small.

In the related works, the paper claims QMIX uses an unweighted mixing whereas the mixing function does weight individual agents’ utilities.


**Q7 Justification For Your Score:**

The paper has some novel contributions, but there are some concerns regarding the relevance of the setting, as well as the experimental evaluation.

**Q9 Complying With Reviewing Instructions:**

1: Yes.

---

### Official Review · Reviewer_KMP7 · 2022-04-14

**Q2(1) Originality/Novelty:** 3
**Q2(2) Significance/Impact:** 3
**Q2(3) Correctness/Technical Quality:** 3
**Q2(6) Clarity Of Writing:** 3
**Q6 Overall Score:** 7
**Q8 Confidence In Your Score:** 3

**Q1 Summary And Contributions:**

In this paper, authors focus on the scalability of multi-agent reinforcement learning and introduce a Dirichlet-multinomial model based
method for maintaining beliefs over the agent population when agents’ actions are not perfectly observable.

**Q2 Assessment Of The Paper:**

More detailed information regarding each of these aspects is given below:

**Q2(4) Quality Of Experiments (Optional):**

3: Good: The experimental evaluation is adequate, and the results convincingly support the main claims.

**Q2(5) Reproducibility:**

2: Fair: Key resources (e.g., proofs, code, data) are unavailable but key details (e.g., proof sketches, experimental setup) are sufficiently well-described for an expert to confidently reproduce the main results.

**Q3 Main Strengths:**

Authors presented new scalable actor-critic MARL algorithms built on the notion of action anonymity that scales polynomially with the number of agents and when compared to recent many-agent MARL methods that assume other agents’ policies or actions are observed accurately, the proposed methods show superior performance in terms of policy quality and convergence speed.

**Q4 Main Weakness:**

1) statistical analysis
2) sharing the code to reproduce the experiments.

**Q5 Detailed Comments To The Authors:**

In general, the manuscript is well written, there is a couple of comments that could improve the readability and the impact of this work: 1) statistical analysis, 2) sharing the code to reproduce the experiments.

**Q7 Justification For Your Score:**

In this work, the authors show that the prediction accuracy of the proposed model improves with more agents and demonstrate scalability of the method on two large domains with up to one hundred agents.

**Q9 Complying With Reviewing Instructions:**

1: Yes.

---

### Decision · Program_Chairs · 2022-05-15

**Decision:**

Accept (Poster)

**Comment:**

Meta Review: This paper receives positive reviews from the reviewers. The major merit, the novelty of the proposed algorithm and its convincing empirical performance, are in particular appreciated by the reviewers.